# Assessment of NUDT5 in Endometrial Carcinoma: Functional Insights, Prognostic and Therapeutic Implications

**DOI:** 10.3390/biomedicines13051136

**Published:** 2025-05-07

**Authors:** Hongfei Yu, Lingling Zu, Yuqin Zang, Fei Teng, Tao Wang, Ming Wu, Yingmei Wang, Fengxia Xue

**Affiliations:** 1Department of Gynecology and Obstetrics, Tianjin Medical University General Hospital, No. 154, Anshan Road, Heping District, Tianjin 300070, China; 2Tianjin Key Laboratory of Female Reproductive Health and Eugenic, Department of Gynecology and Obstetrics, Tianjin Medical University General Hospital, No. 154, Anshan Road, Heping District, Tianjin 300070, China; 3Tianjin Key Laboratory of Lung Cancer Metastasis and Tumor Microenvironment, Tianjin Lung Cancer Institute, Tianjin Medical University General Hospital, No. 154, Anshan Road, Heping District, Tianjin 300070, China

**Keywords:** endometrial carcinoma, *NUDT5*, prognostic factor, PI3K-AKT pathway, tumor progression

## Abstract

**Background**: Endometrial carcinoma (EC) is the most common gynecological malignancy, with increasing incidence contributing to a significant global health burden. Despite recent advancements, the molecular mechanisms underlying EC progression remain insufficiently understood, limiting the development of targeted therapies. This study aims to investigate the role of *nucleoside diphosphate-linked moiety X motif 5 (NUDT5)* in EC and evaluate its potential as a biomarker and therapeutic target. **Methods**: This study analyzed gene expression data from The Cancer Genome Atlas and performed tissue microarray validation to assess *NUDT5* expression in EC samples. Immunohistochemistry was used to evaluate NUDT5 protein levels and their correlation with clinicopathological features. Functional assays, including cell proliferation, migration, invasion, and apoptosis analysis, were conducted to determine the oncogenic effects of NUDT5 in vitro. Weighted gene co-expression network analysis (WGCNA) and experimental validation were performed to explore the impact of NUDT5 on the PI3K-AKT signaling pathway, while tumor growth assays in xenograft models assessed the therapeutic potential of NUDT5 inhibition in vivo. **Results**: *NUDT5* was significantly overexpressed in EC tissues and correlated with advanced histological grade and poor prognosis. Functional experiments demonstrated that NUDT5 promotes cell proliferation, migration, and invasion while inhibiting apoptosis. Mechanistically, NUDT5 activated the PI3K-AKT pathway, contributing to tumor progression. In vivo, *NUDT5* knockdown suppressed tumor growth. **Conclusions**: These findings suggest that *NUDT5* functions as an oncogene in EC, serving as a potential diagnostic and prognostic biomarker. Targeting NUDT5 may provide a novel therapeutic strategy for EC management.

## 1. Introduction

Endometrial carcinoma (EC) is the most prevalent gynecological malignancy worldwide, with approximately 417,000 new cases diagnosed annually [1,2]. Advances in surgical resection, radiotherapy, and chemotherapy have improved outcomes for early-stage EC. However, the prognosis for patients with advanced or recurrent disease remains poor due to the lack of effective targeted therapies [3,4,5]. This gap highlights the critical need to identify molecular biomarkers and therapeutic targets to enhance diagnostic accuracy, prognostic assessment, and treatment outcomes.

Among the molecular pathways implicated in EC, the PI3K-AKT signaling pathway plays a central role in tumorigenesis. Aberrant activation of AKT drives cell cycle progression and proliferation by upregulating key proteins such as Cyclin D1 [6,7,8]. Despite its therapeutic potential, clinical application of AKT inhibitors has been constrained by significant adverse effects, emphasizing the need to identify upstream regulators of AKT activation as alternative targets [9].

Nucleoside Diphosphate-Linked Moiety X Motif 5 (NUDT5) is a hydrolase involved in maintaining nucleotide homeostasis and regulating cellular processes such as transcription elongation and DNA repair. Structurally, NUDT5 contains a conserved Nudix hydrolase domain critical for its enzymatic function [10]. Previous research demonstrated that NUDT5 plays a crucial role in preventing oxidative DNA damage in TNBC cells [11]. Recent research has also suggested a potential link between NUDT5 and hormone-driven cancers, such as breast cancer [12]. Furthermore, a study utilizing whole-exome sequencing identified NUDT5 mutations in EC patients, suggesting that NUDT5 may be associated with the development and progression of EC [13]. However, the precise molecular mechanisms by which NUDT5 interacts with signaling pathways in EC have yet to be fully elucidated. Investigating NUDT5’s function in EC could provide new insights into the molecular mechanisms driving tumor progression and identify therapeutic opportunities.

In this study, we aimed to investigate the expression patterns, clinical relevance, and functional role of NUDT5 in EC. By integrating analyses of public datasets with tissue microarray validation, we assessed the association between NUDT5 expression and clinical outcomes. Additionally, we explored the molecular mechanisms underlying NUDT5-mediated tumorigenesis, focusing on its interaction with the PI3K-AKT signaling pathway. Our findings establish NUDT5 as a key oncogene and therapeutic target in EC, paving the way for novel diagnostic and treatment strategies.

## 2. Materials and Methods

### 2.1. Public Datasets and Bioinformatics Analysis

Gene expression data for endometrial cancer (EC) were retrieved from The Cancer Genome Atlas (TCGA) (https://portal.gdc.cancer.gov) and analyzed using R (version 4.3.1). Differential gene expression analysis was performed using the ‘DESeq2’ package (version 1.40.2). The cutoff thresholds for significance were set as |Fold Change| > 1 and *p*-value < 0.05. To account for multiple testing, False Discovery Rate was controlled using the Benjamini–Hochberg method, with an FDR threshold of 0.05. Gene Ontology (GO) analysis was conducted using the ‘clusterProfiler’ package, and survival analysis was performed using the ‘survminer’ package. Forest plots were generated with the ‘forestplot’ package. Gene set scoring was performed using the GSWA package, which informed subsequent weighted gene co-expression network analysis (WGCNA). Hub genes from the turquoise module were further analyzed for KEGG (Kyoto Encyclopedia of Genes and Genomes) and GO enrichment.

Protein expression data were sourced from the UALCAN database (http://ualcan.path.uab.edu/) using CPTAC data to compare NUDT5 expression between tumor and corresponding normal tissues. Z-values were used for statistical evaluation. Gene Set Enrichment Analysis (GSEA) was performed using the MSigDB dataset (https://www.gsea-msigdb.org/gsea/msigdb), with the ‘c2.cp.all.v2022.1.Hs.symbols.gmt’ dataset used to analyze the enrichment of differentially expressed genes (DEGs) between high and low NUDT5 expression groups. Enrichment analysis for KEGG and GO was conducted using the ‘clusterProfiler’ package, with |log2Fold Change| > 1 and *p* < 0.05 as thresholds.

The Human Protein Atlas (https://www.proteinatlas.org/) provides protein expression data and immunohistochemical staining images of normal and tumor tissues. NUDT5 protein expression was examined using the immunohistochemical data from the Human Protein Atlas.

### 2.2. Tissue Microarray

NUDT5 expression in EC tissues was evaluated using tissue microarrays, which included 276 EC tissue samples and 240 control samples from non-cancerous uterine tissues. Data were obtained from the Department of Obstetrics and Gynecology at Tianjin Medical University General Hospital, including patients who underwent surgery between January 2003 and December 2012. Inclusion criteria required pathological confirmation by at least two independent pathologists, and patients had no prior history of radiotherapy, chemotherapy, or other related adjuvant therapies before surgery. Exclusion criteria included concurrent malignant tumors in other parts of the body and incomplete medical records. All tissue samples used for the microarray were fresh-frozen to preserve the integrity of RNA, DNA, and proteins. The tissue microarray was constructed using standard frozen tissue core technology, where multiple tissue cores were included in each core block for high-throughput analysis. All experiments involving human subjects were conducted with the written consent of each participant. The study adhered to the Declaration of Helsinki and was approved by the Ethics Committee of Tianjin Medical University General Hospital (Ethics code: IRB-2019-KY-130).

### 2.3. Immunohistochemistry

Immunohistochemical staining was performed using the Leica BOND RX system (Leica Biosystems, Wetzlar, Germany). Tissue sections were dewaxed with Bond Dewax Solution and rehydrated. Antigen retrieval was performed with Leica Bond Epitope Retrieval Solution 2 for 20 min. Immunostaining was conducted using Bond Polymer Refine Detection, followed by incubation with primary antibodies for 15 min and secondary antibodies for 8 min. The primary antibody used was rabbit monoclonal anti-NUDT5 antibody(Abcam, ab129172, (Cambridge, UK)). The secondary antibody was HRP-conjugated goat anti-rabbit IgG (Abcam, ab6721). After washing, DAB staining was performed for 10 min, followed by nuclear staining and mounting.

The IRS scoring system was used to evaluate immunostaining intensity and cell positivity rate. The final NUDT5 score was calculated by multiplying the staining intensity score by the cell positivity rate score. The staining intensity was scored as follows: 0, no staining; 1, pale yellow; 2, brown-yellow; 3, brownish brown. The percentage of positively stained cells was scored as follows: 0–5% (0), 6–25% (1), 26–50% (2), 51–75% (3), 76–100% (4).

### 2.4. Cell Culture and Transfection

EC cell lines were purchased from ATCC (Manassas, VA, USA): HEC-1-A (RRID: CVCL_0293), HEC-1-B (RRID: CVCL_0294), ECC-1 (RRID: CVCL_7260), and Ishikawa (RRID: CVCL_2529). HEC-1-A cells were cultured in McCoy’s 5A medium, while HEC-1-B, ECC-1, and Ishikawa cells were cultured in RPMI 1640 medium supplemented with 10% fetal bovine serum (FBS). All cell lines were authenticated within the past three years using short tandem repeat profiling every six months. Mycoplasma testing was regularly performed. The primers for NUDT5 overexpression were as follows: forward, AGGTCGACTCTAGAGGATCCCGCCACCATGGAGAGCCAAGAACCAACGG; reverse, TCCTTGTAGTCCATACCAAATTTCAAG. Transfection efficiency was determined by western blot (WB) and quantitative real-time PCR. In addition, TH5427, a validated NUDT5-targeting inhibitor, was used to treat shCtrl cells to assess the effects of NUDT5 inhibition. TH5427 (15 μM) and the AKT inhibitor Perifosine (5 μM) were applied to EC cells following the manufacturer’s instructions. The experimental groups in this study were as follows: shCtrl (empty plasmid control), SH1 (NUDT5 knockdown group 1), SH2 (NUDT5 knockdown group 2), TH5427 (TH5427-treated shCtrl cells), Vector (empty plasmid overexpression control), OE (NUDT5 overexpression group), and OE treated with an AKT inhibitor.

### 2.5. Western Blot

Cells were lysed in RIPA buffer supplemented with 1% phosphatase inhibitors and phenylmethylsulfonyl fluoride. Total protein was extracted and denatured in loading buffer for 15 min. A sample of 40 μg was loaded onto a 10% SDS-PAGE gel, separated, and transferred to a PVDF membrane (Leica Biosystems, Wetzlar, Germany). The membrane was blocked with 5% milk and incubated overnight with primary antibodies from Abcam (Cambridge, UK) and Cell Signaling Technology (Danvers, MA, USA). The following primary antibodies were used: NUDT5 (Abcam, ab129172), PI3 Kinase (Cell Signaling Technology, 4249), AKT (Cell Signaling Technology, 9272S), phospho-AKT (Cell Signaling Technology, 4060S), E-Cadherin (Cell Signaling Technology, 14472), N-Cadherin (Cell Signaling Technology, 14215S), Vimentin (Cell Signaling Technology, 5741), Caspase-3 (Cell Signaling Technology, 9662), Cyclin D1 (Cell Signaling Technology, 2922S), and β-actin (Abcam, ab8226) as the loading control. After washing with TBST, the membranes were incubated with secondary antibodies for two hours and visualized using ECL detection reagents. Quantification of WB bands was performed using ImageJ (version 1.53t, National Institutes of Health, Bethesda, MD, USA) densitometry analysis. The intensity of the target protein bands was normalized to the β-actin loading control to ensure accurate and consistent results. To ensure the reliability and consistency of the results, all experiments were conducted in three independent replicates.

### 2.6. Cell Proliferation Assays

EC cells were seeded in 96-well plates at a density of 500 cells per well. A total of 110 μL of CCK-8 working solution was added to each well according to the manufacturer’s instructions, and the cells were incubated at 37 °C for 2 h. Absorbance at 450 nm was measured using a microplate reader. For the colony formation assay, 500 cells were seeded in six-well plates and cultured for 14 days. Cells were fixed with 4% paraformaldehyde, stained with crystal violet, and observed under a microscope. Colonies containing more than 10 cells were counted and analyzed using ImageJ software (version 1.53t, National Institutes of Health, Bethesda, MD, USA). All experiments were performed in triplicate to ensure the reliability and consistency of the results.

### 2.7. Transwell Experiment

EC cells (300,000 cells) were seeded into Transwell chambers with or without Matrigel. Culture medium containing 10% FBS was added to the lower chambers. After 24 h of incubation, cells were fixed with 4% paraformaldehyde, and non-invading cells were removed. Invading cells were observed and counted under an inverted optical microscope and analyzed using ImageJ. All experiments were performed in triplicate to ensure result consistency.

### 2.8. Flow Cytometric Analysis

EC cells were seeded in six-well plates and incubated for 24 h. Cells were collected and processed according to the manufacturer’s instructions for apoptosis and cell cycle assays. Flow cytometric analysis was performed using Modfit 5.0 software. Each experiment was repeated three times to confirm the reliability of the results.

### 2.9. mRNA Sequencing

To investigate transcriptional changes following NUDT5 knockdown in HEC-1B cells, total mRNA was extracted and sent to Genewiz (Suzhou, China) for sequencing and preliminary analysis. R (version 4.3.1) was used for further data analysis and visualization, following methods described in the previous section.

### 2.10. Xenograft Nude Mouse Model

BALB/c-nu female nude mice were obtained from the Institute of Hematology, Chinese Academy of Sciences, and randomly assigned to six groups, with 10 mice per group (*n* = 10 per group, total *n* = 60): shCtrl (empty plasmid control), SH1 (NUDT5 knockdown group 1), SH2 (NUDT5 knockdown group 2), Vector (empty plasmid overexpression control), OE (NUDT5 overexpression group), and OE treated with the AKT inhibitor Perifosine. The sample size was determined based on previous literature and feasibility considerations to ensure sufficient statistical power. The animals were randomly assigned to treatment groups.

To establish xenograft models, 1 × 10⁶ HEC-1B cells were subcutaneously injected into the mice. Tumor growth and health were monitored daily. On day 7 post-injection, the AKT inhibitor group received Perifosine (30 mg/kg/day) orally, while the OE-NC and OE groups received equivalent volumes of saline. On day 30, mice were euthanized, and tumor volumes were calculated using the formula (Length × Width^2^)/2. Tumors were excised, photographed, and analyzed by immunohistochemistry.

All animal experiments were conducted in accordance with the guidelines set by the Animal Care and Use Committee of Tianjin Medical University (Ethics code: SYXK2016-0012).

### 2.11. Statistical Analysis

Experiments were performed in triplicate, and data are presented as mean ± SD. Statistical analyses were conducted using GraphPad Prism 8 (GraphPad Software, San Diego, CA, USA). Comparisons between groups were performed using two-tailed Student’s *t*-test for two groups, or one-way ANOVA followed by Tukey’s post hoc test for multiple groups. *p* < 0.05 was considered statistically significant.

## 3. Results

### 3.1. NUDT5 Was Upregulated in Multiple Tumors and High Expression of NUDT5 Predicts Poor Prognosis in Endometrial Carcinoma

To identify potential causative genes in endometrial carcinoma (EC), we analyzed differential gene expression patterns between EC tissues and normal tissues using RNA-Seq data from the TCGA database. A total of 516 upregulated and 495 downregulated differentially expressed genes (DEGs) were identified (Figure 1A). Cox regression analysis revealed eight prognosis-related genes (Figure 1B). Among these, NUDT5, along with TFAP2D and TMEM98, demonstrated strong potential for predicting survival (Figure 1B); however, TFAP2D and TMEM98 did not show significant differential expression (Appendix A). As a result, NUDT5 was identified and further highlighted as a gene potentially involved in cancer development, based on Open Targets analysis (Figure 1C). Next, NUDT5 expression across pan-cancer and normal tissues was examined. The results revealed that NUDT5 was significantly overexpressed at both mRNA and protein levels in multiple cancers, including EC (Figure 1D–F). Kaplan–Meier survival analysis showed that elevated NUDT5 expression was associated with poorer survival outcomes in EC and renal cell carcinoma patients (Figure 1I and Appendix A). Focusing on the role of NUDT5 in EC, we found that upregulated NUDT5 mRNA and protein expression levels correlated positively with more advanced histological grades, suggesting a more aggressive tumor phenotype (Figure 1G–H and Appendix A). Receiver operating characteristic (ROC) curve analysis confirmed that NUDT5 could serve as a potential diagnostic marker for EC, with an area under the curve (AUC) of 0.932, sensitivity of 0.842, and specificity of 0.917 (Figure 1J).

We further analyzed the expression of NUDT5 in endometrial cancer (EC) cell lines and our tissue microarrays. Our findings revealed that NUDT5 was significantly upregulated in high-grade EC cell lines (HEC-1A and HEC-1B) compared to low-grade cell lines (ECC1 and ISHIKAWA) (Figure 1K,L). Consistent with our findings, tissue microarray analysis revealed significantly higher immunohistochemical staining scores in EC tissues compared to control tissues (mean score: 8.16 ± 3.16 for EC tissues vs. 4.31 ± 1.96 for control tissues; *p* < 0.001) (Figure 1M,N).

ROC analysis determined a staining score cut-off of 7 for NUDT5, effectively distinguishing normal from malignant endometrial tissues, with an area under the curve (AUC) of 0.648, sensitivity of 0.721, and specificity of 0.495 (Appendix A). Importantly, risk factor analysis identified a high NUDT5 staining score as an independent risk factor for EC development (Appendix A). Elevated NUDT5 protein expression was significantly associated with poorer overall survival in EC patients, which confirms the conclusions drawn from the analysis of public datasets (Appendix A).

### 3.2. NUDT5 Promotes Cell Proliferation and Inhibits Apoptosis of EC Cells In Vitro

To explore the biological function of NUDT5 in EC cells, we performed enrichment analysis of NUDT5-related DEGs based on the TCGA database and our data. In the GO analysis, NUDT5-related DEGs were primarily enriched in cell proliferation, DNA replication, apoptosis, and cell cycle progression (Figure 2A). KEGG enrichment highlighted NUDT5’s role in modulating cell proliferation, apoptosis, and the AKT pathway (Figure 2A).

To investigate the functional role of NUDT5 in EC cell proliferation and apoptosis, we conducted in vitro experiments on HEC-1A and HEC-1B cells. NUDT5 expression was manipulated through knockdown and overexpression in these cell lines (Figure 2B). Additionally, TH5427, a specific inhibitor of NUDT5, was employed in the experiments. The impact of NUDT5 on cell proliferation was evaluated using the CCK-8 assay and colony formation assays, which showed a marked decrease in EC cell proliferation following the suppression of NUDT5 (Figure 2C,D). Flow cytometry analyses revealed that NUDT5 knockdown significantly hindered the G1 to S phase transition and induced increased apoptosis (Figure 2E,F). Conversely, overexpression of NUDT5 promoted EC cell proliferation, facilitated this cell cycle transition and reduced apoptosis (Figure 2C–F). Gene set enrichment analysis (GSEA) further supported NUDT5’s critical role in the regulation of the cell cycle and apoptosis of EC cells (Figure 2G).

Subsequently, the expression levels of marker proteins associated with the cell cycle and apoptosis were tested. NUDT5 inhibition led to a significant decrease in pCDK4 and Cyclin D1, which are essential for cell cycle progression, while Caspase-3, a key apoptosis regulator, was significantly elevated (Figure 2H). These data demonstrated NUDT5’s essential role in modulating cell proliferation, cell cycle progression, and apoptosis in EC cells.

### 3.3. NUDT5 Enhances the Aggressive Behaviors of EC Cells In Vitro

In addition to proliferation and apoptosis, functional enrichment analysis has also revealed NUDT5’s involvement in the epithelial–mesenchymal transition (EMT) processes of EC. GO enrichment analysis highlighted cell–substrate adhesion and cell shape regulation, while KEGG pathway analysis revealed significant enrichment in adherens junctions and focal adhesions (Figure 3A). GSEA supported NUDT5’s role in crucial biological processes such as cadherin binding and microtubule cytoskeleton organization, which are vital for cellular architecture and migration (Figure 3B). Transwell assays confirmed that NUDT5 facilitates EC cell migration and invasion; EMT behavior was notably enhanced by NUDT5 overexpression and reduced by its suppression (Figure 3C,D). Subsequently, the expression of EMT markers was tested following suppression of NUDT5. Notably, the expression of E-cadherin, an epithelial marker, was significantly increased, while the expressions of N-cadherin and vimentin, both mesenchymal markers, were decreased (Figure 3E). These results support NUDT5 as a key regulator of EMT and its significant impact on the metastasis of EC cells.

### 3.4. Knockdown of NUDT5 Suppressed EC Tumor Growth In Vivo

To assess the therapeutic potential of NUDT5 in vivo, we constructed a xenograft model of EC in nude mice with knockdown of NUDT5 in HEC-1B cells (Figure 4A,B). We observed that silencing NUDT5 resulted in significantly reduced tumor growth, with decreased tumor size and weight compared to the control group (Figure 4C–E and Appendix A). Immunohistochemical analysis further validated significantly reduced proliferation in the NUDT5 knockdown group compared to controls, with lower expression of the proliferation marker Ki-67. Additionally, building on the involvement of the AKT pathway and cell cycle from our previous enrichment analysis, we tested the marker proteins of this pathway; the expressions of pAKT and Cyclin D1 were significantly decreased, suggesting that knockdown of NUDT5 suppressed tumor cell proliferation in vivo potentially through the deactivation of the AKT pathway (Figure 4F).

### 3.5. NUDT5 Modulates the PI3K-AKT Pathway Through a Gene Network

To further explore the potential downstream signaling mechanisms responsible for the tumorigenesis effects of NUDT5 in EC, we employed weighted gene co-expression network analysis (WGCNA) to dissect the gene network modulated by NUDT5 in EC. Setting an appropriate soft threshold ensured that only strong correlations between genes were considered (Figure 5A). Five significant modules were identified, with the turquoise module being particularly significant (Figure 5B–D). Hub genes within the turquoise module were identified and overlapped with the DEGs obtained from our transcriptome sequencing data (Figure 5E). The 348 intersected genes were then subjected to GO and KEGG pathway enrichment analysis. The GO enrichments revealed that NUDT5 plays a crucial role in regulating important biological processes involved in EC, such as extracellular matrix organization, epithelial cell migration, and focal adhesion (Figure 5F). Additionally, KEGG regulatory network analysis highlighted the involvement of significant pathways including the PI3K-AKT pathway, focal adhesion, and cell adhesion molecules, which are known to be crucial for cancer development and progression (Figure 5G). Subsequently, we tested the marker proteins of the AKT pathway, and the results showed that knocking down NUDT5 significantly suppressed the expression of PI3K and AKT proteins, compared to the control group (Figure 5H). These data indicate that NUDT5 plays a vital role in activation the PI3K-AKT pathway and in EC development.

### 3.6. AKT Inhibition Reversed the Biological Effects of NUDT5 In Vitro

To investigate the potential interaction between NUDT5 and the AKT signaling pathway, in vitro experiments were conducted using Perifosine, an AKT inhibitor. Initially, different concentrations of Perifosine were tested to determine an effective dosage. Based on preliminary observations, 5 μM Perifosine was selected for further experiments, as it significantly inhibited EC cell proliferation. EC cells overexpressing NUDT5 were treated with Perifosine, an AKT inhibitor. Functional assays demonstrated a reversal of enhancements in cell proliferation, invasive capabilities, G1/S cell cycle transition, and a decrease in apoptosis induced by NUDT5, suggesting that AKT may be a downstream target of NUDT5 (Figure 6A–E). Additionally, inhibition of AKT had the opposite effect on changes in EMT, caspase-3 activity, and AKT pathway alterations induced by NUDT5 (Figure 6F). Importantly, despite the inhibition of the AKT pathway, NUDT5 protein expression levels remained stable (Figure 6F). These results suggest that NUDT5 may regulate EC development through the activation of the AKT pathway.

### 3.7. NUDT5 Promotes EC Development and AKT Inhibition Disrupted the Effects of NUDT5 In Vivo

Next, to investigate the role of upregulated NUDT5 on EC in vivo, we established a xenograft tumor model with HEC-1B cells overexpressing NUDT5 (Figure 7A). We observed that the overexpression of NUDT5 significantly promoted tumor growth (Figure 7B–E). Furthermore, treating nude mice bearing NUDT5-overexpressed xenograft tumors with Perifosine reversed these effects, indicating that inhibition of the AKT counteracts NUDT5-induced tumor growth (Figure 7B–E). Immunohistochemical analysis confirmed that protein levels of Ki-67, pAKT, and Cyclin D1, which were significantly increased due to NUDT5 overexpression, decreased after Perifosine treatment, effectively deactivating the AKT/Cyclin D1 signaling pathway (Figure 7F). These results indicate that NUDT5 promotes EC tumor growth by activating the AKT pathway.

## 4. Discussion

In the present study, we investigated the clinical relevance and biological function of NUDT5 in endometrial carcinoma (EC). We analyzed both publicly available datasets and our own tissue microarray data to explore the association between NUDT5 expression and clinicopathological features, as well as patient survival outcomes. We also performed functional experiments to assess the biological role of NUDT5 in EC cell lines. Our results indicate that NUDT5 is significantly upregulated in EC tissues, and its expression correlates with poorer clinical outcomes, highlighting its potential as a prognostic marker and therapeutic target in EC.

Our study confirmed the overexpression of NUDT5 at both the mRNA and protein levels in EC tissues compared to normal endometrial tissues. Tissue microarray analysis revealed a significant association between high NUDT5 expression and higher histological grades of EC, consistent with findings from public data analysis and earlier clinical studies in other cancers [14,15,16]. Importantly, our Kaplan–Meier survival analysis demonstrated that elevated NUDT5 expression is associated with poorer overall survival, underscoring its potential as a prognostic marker in EC. These observations highlight the importance of integrating both transcriptomic and proteomic data when evaluating the clinical significance of potential biomarkers, as protein expression may provide more direct insights into tumor biology and prognosis.

Functional analyses revealed that NUDT5 plays a critical role in promoting EC cell proliferation, survival, and migration. Knockdown of NUDT5 in EC cell lines led to a marked reduction in cell proliferation, with flow cytometry analysis showing an arrest at the G1 phase and increased apoptosis. These results suggest that NUDT5 contributes to cell cycle regulation and apoptosis evasion, which are crucial processes for tumor growth and resistance to cell death. Moreover, NUDT5 overexpression enhanced EC cell migration and invasion, indicating its role in promoting tumor metastasis. These findings are partially consistent with previous studies showing that NUDT5 functions as an oncogene in other cancers by modulating key processes involved in tumor progression [17,18]. Our data also suggest that NUDT5 may influence epithelial–mesenchymal transition, a critical step in tumor metastasis. The upregulation of mesenchymal markers and the downregulation of epithelial markers in NUDT5-overexpressing EC cells further support its role in enhancing metastatic potential. These results suggest that NUDT5 could be a central player in the aggressive behavior of EC and may contribute to therapeutic resistance by promoting cellular plasticity and metastasis.

In earlier studies, NUDT5 has been implicated in ADP-ribose metabolism, which plays a key role in maintaining genomic stability and DNA damage response, processes that are crucial for tumor progression [12]. Recent studies suggest that NUDT5 might facilitate cancer cell survival by maintaining high levels of nucleotide pools, which help cells cope with increased replication stress and DNA damage [15]. Moreover, our pathway enrichment analysis identified NUDT5-associated genes involved in PI3K-AKT signaling, and further validation experiments demonstrated that NUDT5 knockdown led to a decrease in *p*-AKT levels, suggesting a potential role in activating this pathway, which is a central regulator of cell survival, growth, and metastasis in EC and other malignancies [19,20]. The role of NUDT5 in modulating PI3K-AKT signaling further underscores its potential as a therapeutic target. This pathway is particularly relevant in cancer therapy, as drugs targeting this pathway, such as AKT inhibitors, have been explored [21,22,23,24,25,26]. However, their clinical application remains limited due to dose-dependent toxicity and off-target effects, such as hyperglycemia, immunosuppression, and gastrointestinal complications [9,18]. In this context, targeting NUDT5 could provide an upstream therapeutic alternative that modulates this pathway and offers a novel approach to overcome the limitations of direct AKT inhibition, highlighting the potential significance of our findings.

Our study has several limitations that must be addressed. First, while we demonstrated the biological function of NUDT5 in EC cell lines, further in vivo studies are needed to confirm its role in tumorigenesis and metastasis. Additionally, further exploration into the molecular mechanisms underlying NUDT5’s regulation of the PI3K-AKT pathway is warranted to better understand its functional relevance and therapeutic potential in EC.

In conclusion, our study demonstrated that NUDT5 is a potential oncogene and prognostic biomarker in endometrial carcinoma. High NUDT5 expression is associated with poor prognosis and aggressive biological behaviors, including increased cell proliferation, migration, and invasion. These findings suggest that NUDT5 could be a potential therapeutic target for restricting tumor progression and metastasis in EC. Further research is needed to explore the underlying molecular mechanisms and to evaluate the therapeutic efficacy of NUDT5-targeted interventions in preclinical and clinical settings.

## Figures and Tables

**Figure 1 biomedicines-13-01136-f001:**
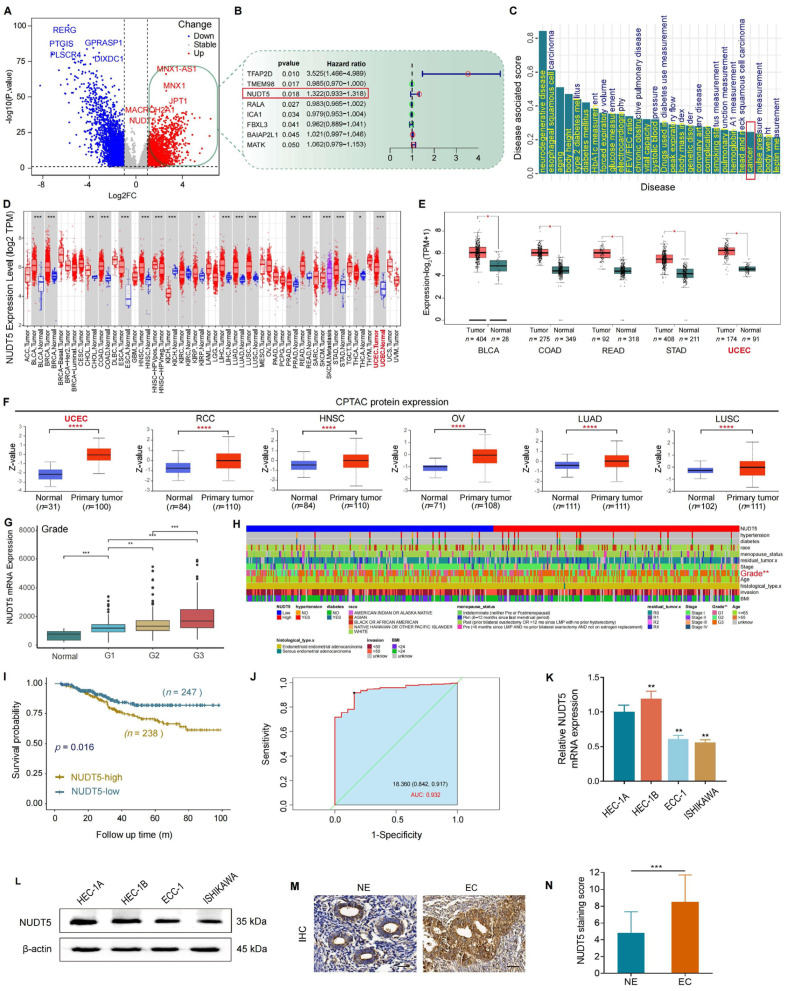
NUDT5 expression in multiple cancers and its clinical and prognostic characteristics in EC. (**A**) Differential expression analysis based on TCGA datasets, visualized by a volcano plot. Red and blue dots indicate significantly upregulated and downregulated genes, respectively. (**B**) Cox regression analysis of representative differential genes, identifying NUDT5 as a prognosis-related gene. (**C**) Analysis of correlation between NUDT5 expression and disease by Open Target Web. (**D**) Overview of pan-cancer NUDT5 mRNA expression. (**E**) Significant upregulation of NUDT5 mRNA expression in multiple cancers compared to normal tissues. (**F**) Significant increase in NUDT5 protein across various cancers according to CPTAC datasets. (**G**) Positive correlation of NUDT5 mRNA expression with EC histological grading. (**H**) Correlation of NUDT5 mRNA expression with clinical characteristics in EC patients, shown as a heatmap. (**I**) NUDT5 mRNA expression and prognosis in patients with EC. (**J**) ROC analysis for NUDT5 mRNA in EC patients: optimal cutoff value of 18.36, AUC of 0.932, sensitivity of 0.842, and specificity of 0.917. (**K**) Differential expression of NUDT5 mRNA and (**L**) protein in high-grade EC cell lines (HEC-1A and HEC-1B) and low-grade EC cell lines (ECC1 and Ishikawa). (**M**) Representative images demonstrating NUDT5 protein expression in EC and normal tissues. (**N**) Statistical analysis of NUDT5 staining scores indicating significantly higher NUDT5 protein expression in EC tissues compared to normal tissues. * *p* < 0.05, ** *p* < 0.01, *** *p* < 0.001, **** *p* < 0.0001.

**Figure 2 biomedicines-13-01136-f002:**
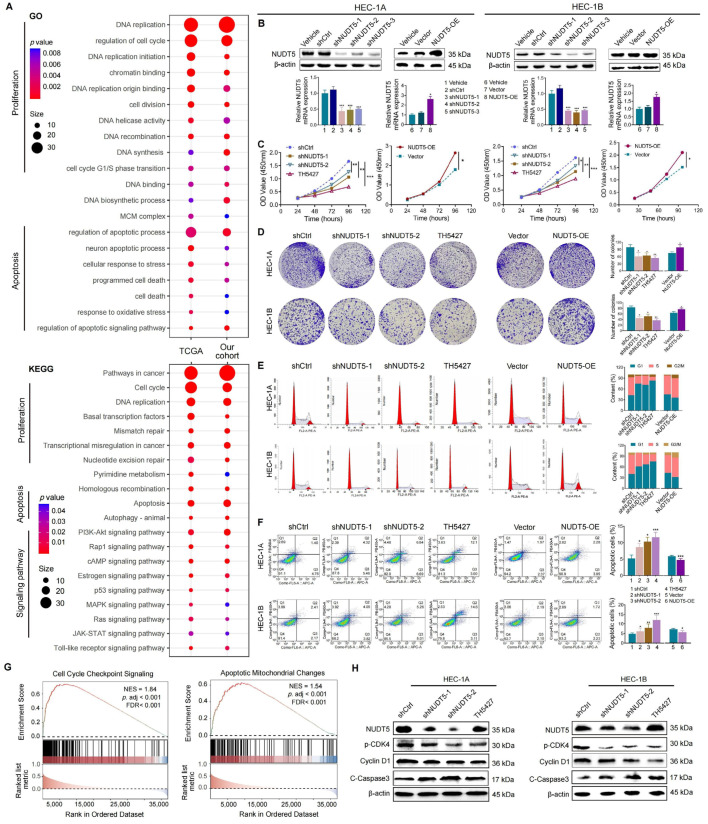
NUDT5 mediates proliferation and apoptosis of EC cells in vitro. (**A**) Intersected GO and KEGG enrichments of NUDT5-related DEGs in TCGA and our cohort, focusing on cellular proliferation and apoptosis. (**B**) NUDT5 was silenced and overexpressed in HEC-1A and HEC-1B EC cell lines. (**C**) Cell proliferation measured by CCK8 assay and (**D**) colony formation assay. (**E**) Cell cycle analysis of the indicated groups. (**F**) Apoptosis levels in the indicated groups. (**G**) GSEA enrichment for NUDT5-related DEGs in TCGA and our cohort, focusing on cell cycle and apoptosis. (**H**) Marker proteins of the cell cycle and apoptosis tested by western blot. * *p* < 0.05, ** *p* < 0.01, *** *p* < 0.001.

**Figure 3 biomedicines-13-01136-f003:**
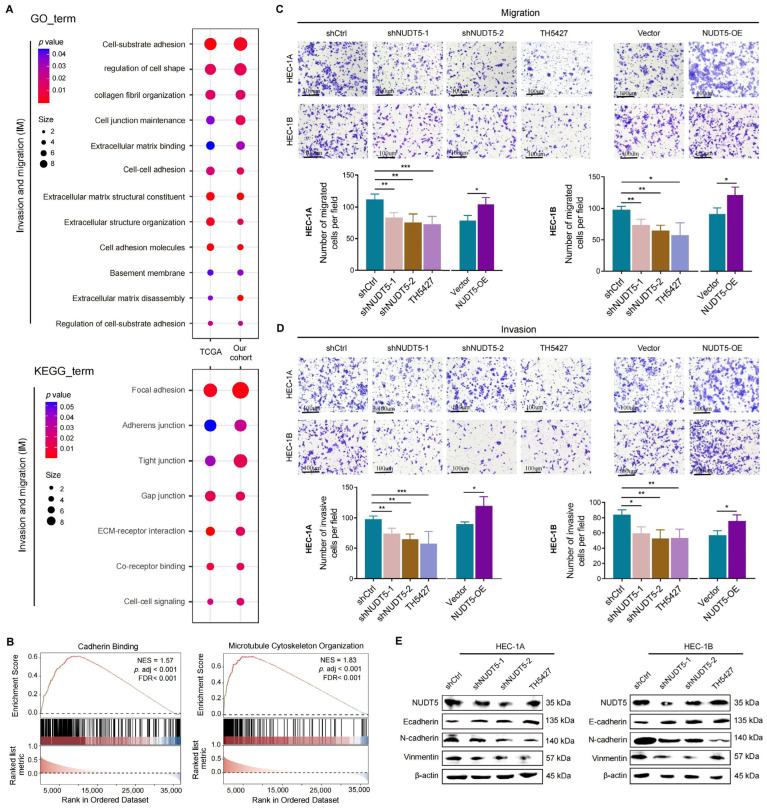
NUDT5 regulates invasion and migration in EC cells. (**A**) Intersected GO and KEGG enrichments, indicating NUDT5 regulates the cellular EMT process. (**B**) GSEA for NUDT5-related DEGs in TCGA and our cohort, focusing on cadherin binding and microtubule cytoskeleton organization. (**C**) Cell migration and (**D**) invasion abilities measured by Transwell assays. (**E**) Marker proteins of the EMT process tested by western blot. * *p* < 0.05, ** *p* < 0.01, *** *p* < 0.001.

**Figure 4 biomedicines-13-01136-f004:**
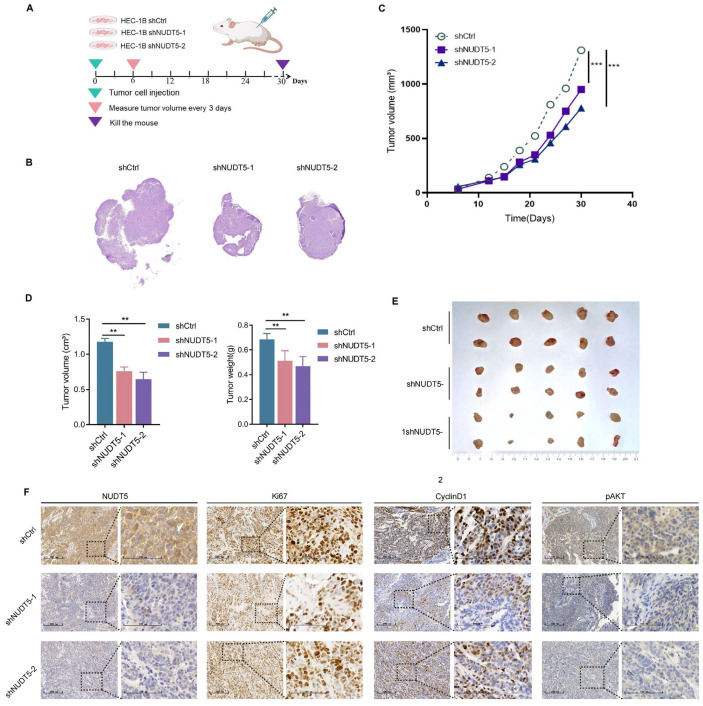
Knockdown of NUDT5 suppresses EC tumor growth in vivo. (**A**) Xenograft model using NUDT5-knockdown HEC-1B cells. (**B**) Overview of H&E-stained microtumor sections. (**C**) Tumor growth curves, *** *p*  <  0.001 vs. shCtrl. (**D**) Effect of NUDT5 knockdown on tumor volume in vivo. (**E**) Body weight and volume in the xenograft tumor model. (**F**) Marker proteins of the cell cycle tested by immunohistochemistry. ** *p* < 0.01, *** *p* < 0.001.

**Figure 5 biomedicines-13-01136-f005:**
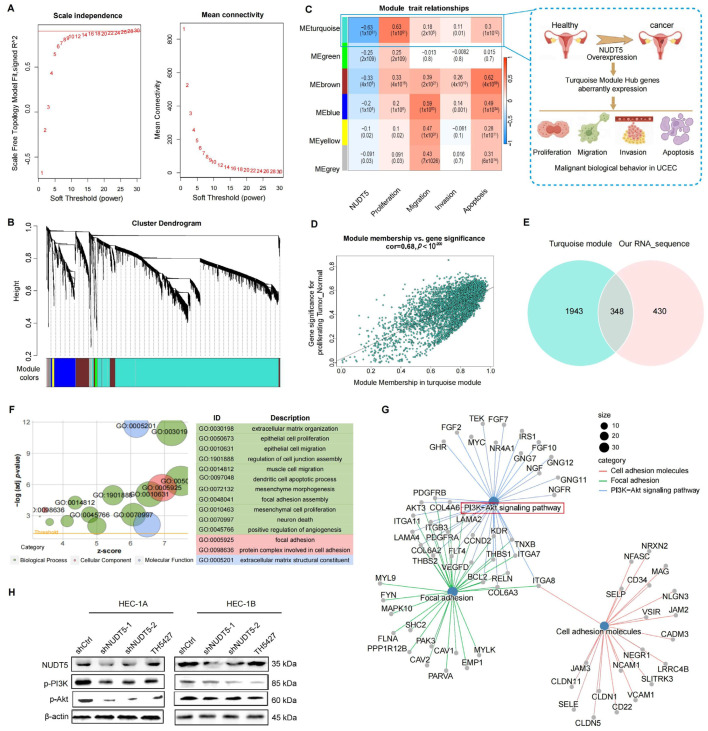
NUDT5-mediated regulatory network of malignant biological behaviors in EC. (**A**) Analysis of appropriate soft-thresholding powers (β) for WGCNA. Analysis of the mean connectivity for various soft-thresholding powers. (**B**) Generation of gene dendrograms and modules within the TCGA cohort, modules were constructed with different colors. (**C**) Correlation analysis between modules and the role of NUDT5 in malignant biological behaviors, illustrated schematically. (**D**) Correlation analysis between the turquoise module and malignant biological behaviors. (**E**) Intersection of hub genes from the turquoise module with DEGs from our cohort. (**F**) GO enrichment analysis of intersected genes. (**G**) KEGG enrichment analysis of intersected genes, identifying the PI3K-AKT pathway. (**H**) Western blot analysis of key proteins in the PI3K-AKT pathway.

**Figure 6 biomedicines-13-01136-f006:**
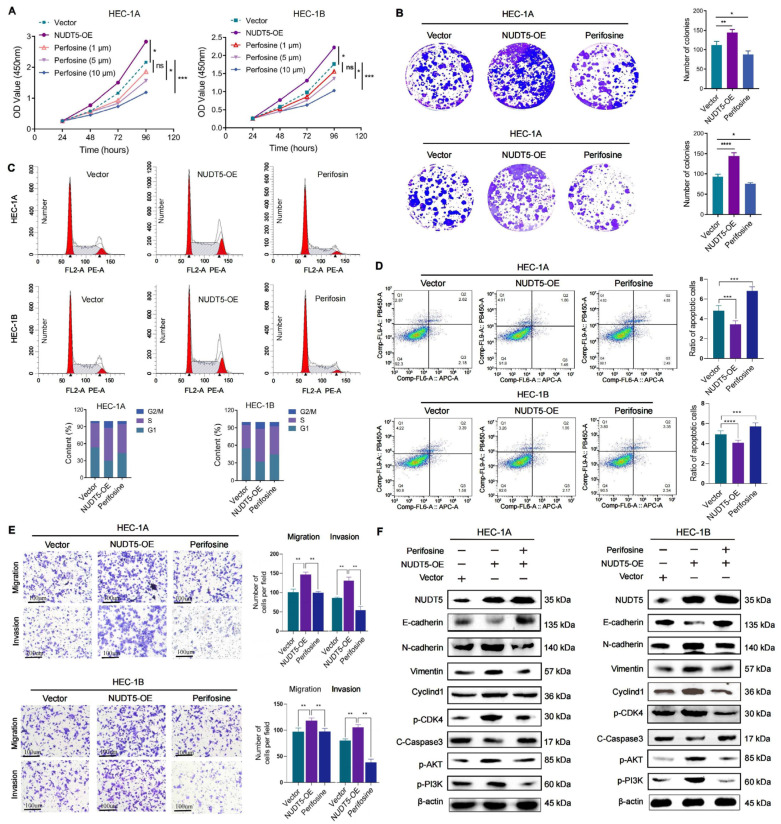
Neutralization of NUDT5’s biological effects by an AKT inhibitor in vitro. (**A**) CCK8 assay illustrating changes between the indicated groups. (**B**) Assessment of cell proliferation through colony formation assays. (**C**) Cell cycle and (**D**) apoptosis levels of indicated groups evaluated by flow cytometry. (**E**) Cell migration and cell invasion capabilities assessed by Transwell assays. (**F**) Analysis of protein levels for EMT-related genes, PI3K-AKT, and CyclinD1 pathways via western blot. Significance levels are indicated as follows: ns (not significant), * *p* < 0.05, ** *p* < 0.01, *** *p* < 0.001, **** *p* < 0.0001.

**Figure 7 biomedicines-13-01136-f007:**
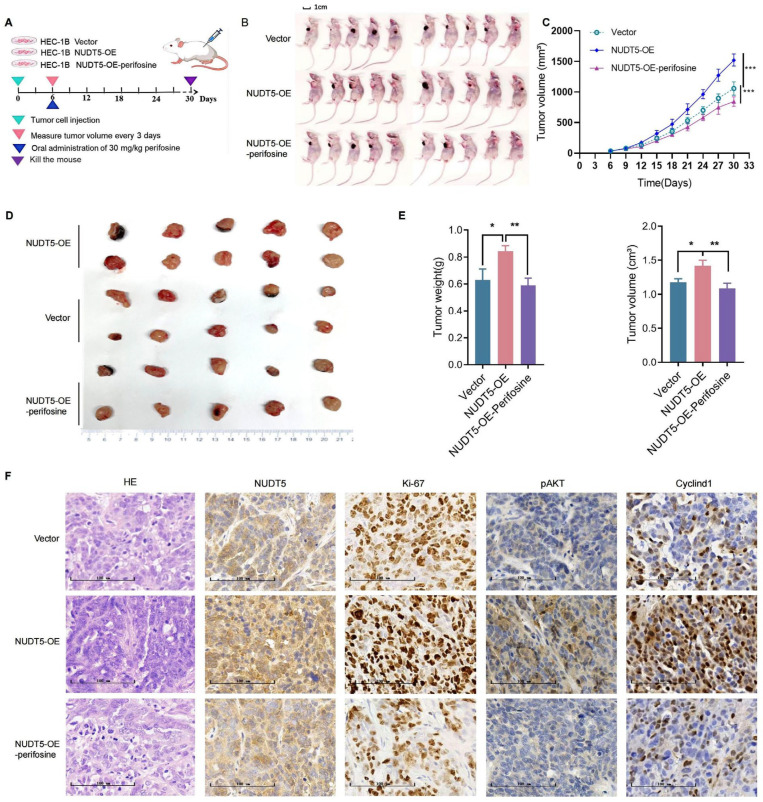
Disruption of NUDT5’s biological effects by an AKT inhibitor in vivo. (**A**) Xenograft model using NUDT5-overexpressing HEC-1B cells. (**B**) At the end of the experiment, the mice were sacrificed, and the tumors were collected. (**C**) Tumor growth curves. *** *p*  <  0.001. (**D**) Representative images of tumor tissues from each group. (**E**) Assessment of body weight (left) and tumor volume (right) at the end of the experiments in the xenograft model. (**F**) Immunohistochemical analysis of NUDT5, Ki-67, pAKT, and Cyclin D1 protein expression in tumor tissues. * *p* < 0.05, ** *p* < 0.01, *** *p* < 0.001.

## Data Availability

The public databases used in the article are derived from TCGA (https://portal.gdc.cancer.gov/), UACAN (https://ualcan.path.uab.edu/), and HPA (The Human Protein Atlas), which can be obtained from the official websites. Other relevant data supporting this research can be reasonably requested from the corresponding author.

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
