# Peer review of "Assessment of NUDT5 in Endometrial Carcinoma: Functional Insights, Prognostic and Therapeutic Implications"

_biomedicines, 2025, doi:10.3390/biomedicines13051136_

Round 1
Reviewer 1 Report
Comments and Suggestions for Authors
Dear authors;
Thank you for your valuable study. But there are some questions and gaps. below I provide my commentaries.
Title:
- I think the title can be improved.
Abstract:
- It is better to add your “materials and methods” and your study design in the abstract.
- which data set did you use?
- In line 6 you wrote “Here, we identified NUDT5 as a novel oncogene in EC”, but there are the other articles in the past years that mentioned this gene as a non-canonical cancer-related gene. For example, you can see this article “Identification of novel mutations in endometrial cancer patients by whole-exome sequencing” that Published online on: March 20, 2017. So, I think it is better to change the “Novel” word or emphasis in this sentence.
Introduction:
- There is a double space in paragraph 2, line 6.
- Also, you can explain the role of NUDT5 in EC in this section.
Materials and Methods:
- Section 2.2: please Add the number of samples and ethics code.
- Please say, how do you collect your sample for microarray? Did you work fresh or freeze?
- Section 2.3: what was your primary and secondary antibodies?
- Generally, how much is your replication in each test? It is be better to say it in the first of M&M or each section, instead of the end of M&M in statistical analyses. It was like a mystery in all sections
- Please add the number of EC cells were seeded in each well?
- Section 2.10: explain each experimental group for the first time and insert their abbreviation in Also, replication repeat should be added here, too. Please make sure to rewrite this section.
Results:
- First, please add aaaaallllllllll p-value and mean±SD in the text.
- 1.M, add the scale bar and for fig. 1.N add the mean±SD. Also, add it in the text.
- What is the difference between fig. 1.I and fig. S2C?
- Section 3.2: Please introduce your groups in the text and figures clearly. What is shNUDT51-3? Are they sham? Or what is OE? You missed this important data. The reader should be able to easily connect with your groups and not constantly search for your meaning.
- Where is your real-time PCR for apoptotic genes?
- Why did not you introduce TH5427 in M&M section?
- It seems that the section 3.3 is not the result of the 2.10 section. You didn’t say anything about NUDT5 knockdown in M&M section or where are your injection groups? Please rewrite them clearly and make a more complete schematic figure.
- The fig. 5H need more explanation. Talk about TH5427 group. It seems that NUDT5 expression in this group is similar to ShNUDT5. Also, what is difference between HEC-1A and HEC-1B group? It seems that p-AKT has more expression in HEC-1B group. In addition, do you have any quantitative data?
- Section 3.6: you didn’t mention perifosine for in vitro section in M&M. however, in figure 6A, I can see you used three dosages for in vitro culture. Why? It is really strange to find new thing in the result.
- 7D and E needs more explanation. Also, the arrangement of groups in the histological image and chart can confuse the reader.
Discussion:
- You discussion is too short. Please add more explanations.
Author Response
Dear Reviewer,
Thank you for your detailed and insightful feedback on our manuscript. We truly appreciate the time you took to review our work and the thoughtful suggestions you provided. We have carefully addressed all of your comments, and below we outline the changes made in the revised manuscript:
Title:
Comments 1: The title could be improved.
Response 1: We have revised the title for better clarity and precision to reflect the key aspects of our study. The updated title can be found on the title page of the manuscript.
Abstract:
Comments 2: It is better to add your “materials and methods” and your study design in the abstract. Which data set did you use? Also, in line 6 you wrote “Here, we identified NUDT5 as a novel oncogene in EC”, but other articles in the past have already mentioned this gene as a non-canonical cancer-related gene. For example, the article “Identification of novel mutations in endometrial cancer patients by whole-exome sequencing” (March 20, 2017). Therefore, the term “novel” should be revised.
Response 2: We have added the “Materials and Methods” section and the study design to the abstract to provide a more comprehensive overview of our approach. Additionally, we have revised the sentence mentioning NUDT5 to acknowledge its identification in previous studies and have removed the word “novel” to ensure accuracy. The updated abstract can be found in the abstract paragraph..
Introduction:
Comments 3: There is a double space in paragraph 2, line 6 of the Introduction.
Response 3: We have corrected the formatting issue and removed the double space in Page 2, Paragraph 2, Line 6.
Comments 4: Explain the role of NUDT5 in EC in the Introduction.
Response 4: We have added an explanation of the role of NUDT5 in endometrial carcinoma (EC) in the Introduction, highlighting its potential as an oncogene and its relevance to the progression of EC. This addition can be found on Page 3, Paragraph 3.
Materials and Methods:
Comments 5: Section 2.2: Please add the number of samples and ethics code. Clarify how you collected your sample for microarray (fresh or frozen)?
Response 5: We have added the number of samples used in the study and included the relevant ethics code in Section 2.2. Additionally, we have clarified that the microarray samples were collected frozen. These details can be found in Section 2.2.
Comments 6: Section 2.3: What were your primary and secondary antibodies? Also, clarify the replication in each test.
Response 6: We have specified the primary and secondary antibodies used in the experiments.Additionally, we have clarified the replication in each experiment in Methods section.
Comments 7: Please add the number of EC cells seeded in each well.
Response 7: We have added the number of EC cells seeded in each well to Section 2.6 and 2.7, as requested.
Comments 8: Section 2.10: Explain each experimental group for the first time and insert their abbreviation. Also, include replication details here. Please make sure to rewrite this section.
Response 8: We have revised Section 2.10 to include a detailed explanation of each experimental group, their abbreviations, and the replication numbers used.
Results:
Comments 9: Please add all p-values and mean±SD in the text, especially for Figure 1.M and 1.N.
Response 9: We have included the p-values and mean ± SD in the relevant sections of the manuscript, especially for Figure 1.M and 1.N, as well as in the figure legends for better clarity.
Comments 10: What is the difference between fig. 1.I and fig. S2C?
Response 10: To clarify, as described in the text, Figure 1.I illustrates the correlation between NUDT5 expression levels and survival prognosis in EC patients based on RNA-Seq data from the TCGA database. On the other hand, Figure S2C presents a similar survival analysis but uses clinical data derived from tissue microarray analysis. The data sources are different. We hope this clarifies the distinction between these two figures.
Comments 11: In Section 3.2, please introduce your groups clearly in the text and figures. What is shNUDT51-3? Are they sham? What is OE?
Response 11: We have clarified the experimental groups and their abbreviations (including shNUDT5 group and OE) in Methods section, making it easier for readers to follow. As outlined in Methods section, the experimental groups in our study are as follows: shCtrl (empty plasmid control), SH1 (NUDT5 knockdown group 1), SH2 (NUDT5 knockdown group 2), TH5427 (TH5427-treated shCtrl cells), Vector (empty plasmid overexpression control), OE (NUDT5 overexpression group), and OE treated with an AKT inhibitor.We hope this resolves the issue and provides a clearer understanding of our experimental design.
Comments 12: Where is your real-time PCR for apoptotic genes?
Response 12: Thank you for your valuable comment. Unfortunately, we did not perform real-time PCR for apoptotic genes in this study. The focus of our research was to investigate the expression of NUDT5 and its potential role in endometrial carcinoma progression. Although apoptosis is a key process in cancer biology, we did not specifically examine apoptotic gene expression via PCR in the current set of experiments.Caspase-3, as mentioned, is a crucial protein in the execution phase of apoptosis, and its activity is primarily regulated at the protein level. While mRNA expression can be assessed by PCR, apoptosis is typically evaluated by analyzing the activation and cleavage of apoptotic proteins using methods like Western blotting. We opted to focus on protein-level assays in this study to better capture the functional aspect of apoptosis.We appreciate your suggestion, and while we did not include this specific analysis in the current work, we will certainly consider incorporating such assays in future studies to provide a more comprehensive understanding of apoptosis in relation to NUDT5.
Comments 13: Why didn’t you introduce TH5427 in the Materials and Methods section?
Response 13: We have added the details regarding TH5427 in the Materials and Methods section, This addition is visible in Section 2.4.
Comments 14: It seems that the section 3.3 is not the result of the 2.10 section. You didn’t say anything about NUDT5 knockdown in M&M section or where are your injection groups? Please rewrite them clearly and make a more complete schematic figure.
Response 14: Thank you for your valuable comments. We acknowledge that the description of the experimental groups in Section 2.10 was not detailed enough, which may have led to confusion regarding the results presented in Section 3.3. To address this issue, we have now revised Section 2.4 (Cell Culture and Transfection) to provide a clearer and more detailed explanation of the experimental grouping, ensuring consistency with the findings in Section 3.3. These modifications will help readers better understand our study design and results.We appreciate your insightful feedback and have carefully revised the manuscript accordingly.
Comments 15: The fig. 5H need more explanation. Talk about TH5427 group. It seems that NUDT5 expression in this group is similar to ShNUDT5. Also, what is difference between HEC-1A and HEC-1B group? It seems that p-AKT has more expression in HEC-1B group. In addition, do you have any quantitative data?
Response 15: Thank you for your insightful comments.Regarding the TH5427 group, although the Western blot results show that NUDT5 expression in this group appears similar to the shNUDT5 group, this is expected because TH5427 primarily functions as an inhibitor of NUDT5 activity rather than affecting its expression levels.We conducted this experiment in two independent EC cell lines, HEC-1A and HEC-1B, to validate the robustness of our findings. The observed difference in p-AKT expression between these two cell lines is likely due to inherent variations in protein expression levels between different cell lines, which is a common phenomenon in biological studies.If needed, we can provide quantitative data to further support our findings. Please let us know if additional details or analyses are required.
Comments 16: Section 3.6, you didn’t mention perifosine for in vitro section in M&M. however, in figure 6A, I can see you used three dosages for in vitro culture. Why? It is really strange to find new thing in the result.
Response 16: Thank you for your valuable feedback. We have revised Section 2.4 to clarify the introduction of Perifosine in the Materials and Methods section (now detailed in Section 2.4). The selection of Perifosine concentrations for in vitro experiments was based on the manufacturer’s instructions and preliminary testing. Specifically, as described in text 5 uM Perifosine was chosen as it significantly inhibited EC cell proliferation and was used as the effective concentration for subsequent experiments. The revised manuscript now explicitly states the rationale for this dosage selection.We appreciate your insightful comments, which have helped us improve the clarity and accuracy of our manuscript.
Comments 17: Figures 7D and E need more explanation. The arrangement of groups in the histological image and chart could confuse the reader.
Response 17: Thank you for your constructive comments. In response to your feedback, we have revised the legend for Figure 7D to clarify the presentation of tumor images. We sincerely appreciate your thoughtful comments, which have significantly contributed to improving the clarity and accuracy of our manuscript.
Discussion:
Comments 18: The discussion is too short. Please add more explanations.
Response 18: We have expanded the Discussion section to include more detailed explanations of our findings, their implications, and potential future directions.
We believe these revisions address all your concerns and improve the manuscript significantly. We hope the revised version meets your expectations, and we look forward to any further feedback.
Thank you again for your valuable comments and suggestions.
Best regards,
Yingmei Wang
Reviewer 2 Report
Comments and Suggestions for Authors
Comments to authors-
The manuscript entitled “NUDT5 Drives Endometrial Carcinoma Progression via Activation of the PI3K-AKT Pathway: Prognostic and Therapeutic Implications” was well written and easy to follow. I have following comments/suggestions
- In the cell cycle experiment, are you aiming to demonstrate an elongated S-phase or an increased number of cells passing through the S-phase? Some samples indicate a shorter S-phase with KD.
- The representative images for the control sample of HEC-1A show very few cells, whereas the treated samples display nearly four times as many. Please update the representative FACS cell cycle control images for better comparison.
- Ensure a consistent scaling criterion across all FACS graphs to facilitate easier interpretation of changes.
- Include a scale bar for the migration and invasion staining sample images in Figure 3.
- The tumor images in Figure 4C are not very clear. Since tumor images are already provided in Figure 4E, consider moving Figure 4C to the supplementary section.
- The representative apoptosis images from FACS do not correlate with the corresponding bar graphs. Please update them accordingly.
- The tumor volume graphs (Figure 7C) are missing error bars. Please correct this.
- Some IHC images appear slightly blurred. Please update them for better clarity.
Author Response
Dear Reviewer,
Thank you for your thoughtful and helpful feedback on our manuscript. We appreciate your detailed suggestions and have carefully addressed each one. Your comments have been invaluable in helping us improve the clarity and quality of our work. Below, we outline the changes we have made in response to your feedback:
Comments 1:
In the cell cycle experiment, are you aiming to demonstrate an elongated S-phase or an increased number of cells passing through the S-phase? Some samples indicate a shorter S-phase with KD.
Response 1:
Thank you for your insightful comments. Our aim was to demonstrate that NUDT5 knockdown (KD) inhibits the G1/S phase transition, leading to more tumor cells being arrested in the G1 phase.
Comments 2:The representative images for the control sample of HEC-1A show very few cells, whereas the treated samples display nearly four times as many. Please update the representative FACS cell cycle control images for better comparison.
Response 2: Thank you for your suggestion. We acknowledge the concern regarding the representative FACS cell cycle control images and have updated Figure 2 to provide a clearer and more accurate comparison between control and treated groups.We appreciate your feedback, which has helped us enhance the clarity and accuracy of our data presentation.
Comments 3:
Ensure a consistent scaling criterion across all FACS graphs to facilitate easier interpretation of changes.
Response 3:
Thank you for your suggestion. We acknowledge the importance of maintaining a consistent scaling criterion across all FACS graphs to facilitate easier interpretation. However, the default scaling in the analysis software varies based on the distribution of events in each sample. Despite this, we have carefully ensured that the data presentation remains accurate and comparable across groups.
Comments 4:
Include a scale bar for the migration and invasion staining sample images in Figure 3.
Response 4:
A scale bar has been added to the migration and invasion staining sample images in Figure 3.
Comments 5:
The tumor images in Figure 4C are not very clear. Since tumor images are already provided in Figure 4E, consider moving Figure 4C to the supplementary section.
Response 5:
We have moved the tumor images from Figure 4C to the supplementary Figure S2D since the images in Figure 4E already provide clear representations. This change can be seen in Figure S2D.
Comments 6:
The representative apoptosis images from FACS do not correlate with the corresponding bar graphs. Please update them accordingly.
Response 6:
Thank you for your valuable feedback. We have re-evaluated the apoptosis data by performing multiple repeated experiments to ensure accuracy. The apoptosis rate was consistently calculated as the sum of early apoptosis (lower right quadrant) and late apoptosis (upper right quadrant). Based on this validation, we have updated the representative FACS images and corresponding bar graphs to ensure consistency. We appreciate your insightful comments, which have helped us improve the accuracy and clarity of our manuscript.
Comments 7:
The tumor volume graphs (Figure 7C) are missing error bars. Please correct this.
Response 7:
We have added error bars to the tumor volume graphs in Figure 7C to improve the data presentation.
Comments 8:
Some IHC images appear slightly blurred. Please update them for better clarity.
Response 8:
Thank you for your suggestion. We have updated the IHC images with higher-resolution versions to improve clarity. Additionally, we have ensured that the image quality is optimized for better visualization. If needed, we are happy to provide the original, uncompressed images for further evaluation. We appreciate your feedback, which has helped us enhance the quality of our manuscript..
We have carefully incorporated all the revisions and believe that the manuscript is now improved both in terms of clarity and presentation. We hope that these changes address your concerns and that the revised manuscript will meet your expectations.
Thank you again for your constructive feedback. We look forward to your further comments.
Best regards,
Yingmei Wang
Reviewer 3 Report
Comments and Suggestions for Authors
NUDT5 Drives Endometrial Carcinoma Progression via Activation of the PI3K-AKT Pathway: Prognostic and Therapeutic Implications
This manuscript Xue et al., provide an in-depth analysis of the role of NUDT5 in endometrial carcinoma, demonstrating its oncogenic potential and its involvement in the PI3K-AKT signaling pathway. The study integrates bioinformatics analysis, in vitro assays, and in vivo models to establish NUDT5 as a potential prognostic marker and therapeutic target. I believe the findings are novel, clinically relevant, and well-supported by experimental data.
Despite these merits, however, several weaknesses must be addressed before publication, particularly in statistical rigor, clarity of methodology, mechanistic interpretation, and discussion of limitations.
For the starters, I think the novelty of the study is not sufficiently emphasized. Several studies have already identified NUDT5 as an oncogenic factor in other cancers (breast, gastric, lung, and head & neck cancers).
A direct comparison with these cancers is missing. How is the role of NUDT5 in EC different?
I’d justify why NUDT5 is a more promising target than existing PI3K-AKT inhibitors, which have significant toxicity concerns.
The cutoff values for differentially expressed genes are not explicitly defined (Fold-change? p-value?).
The statistical corrections for multiple testing (e.g., FDR, Bonferroni) are missing. I wonder were the false positives accounted for?
Overexpression and knockdown are performed, but rescue experiments or complementation exp or re-expressing NUDT5 after knockdown are missing. These are essential to confirm that the effects are specifically due to NUDT5 and not off-target effects.
Western blot normalization missing. Was loading control (b-actin or GAPDH) quantified for WB bands? Quantification of WB bands with, ImageJ densitometry analysis is missing.
In the apoptosis assays, I’d report Annexin V/PI staining results as bar graphs with statistical significance.
Additionally, sample size for in vivo experiments is not reported. were the number of mice per group statistically justified?
Tumor growth curves lack error bars. these are necessary for statistical significance testing.
Body weight monitoring and toxicity assessment are missing. was NUDT5 inhibition associated with any side effects?
H&E staining of tumors should be included to validate histopathological features.
I’d add line and page number so as to make it easier to comments and suggestions. Please check out sentence referenced with ref no 11.
Section 2.4 check for line, spacing etc
Please mention full forms whenever required for example, Pmsfsection 2.6, why 110u of solution was used?
Good luck and keep up the good work,
Cheers
Author Response
Dear Reviewer,
Thank you for your thorough and thoughtful review of our manuscript. We truly appreciate your positive remarks about the study’s novelty and clinical relevance, as well as your constructive comments. We have carefully addressed each of your concerns, and below we outline the changes and clarifications we have made in the revised manuscript:
Comments 1: The novelty of the study is not sufficiently emphasized, and a direct comparison with other cancers where NUDT5 has been identified as an oncogenic factor is missing. How is the role of NUDT5 in EC different?
Response 1: Thank you for your insightful feedback. To address your concerns, we have revised the Introduction section to better emphasize the novelty of our study. We hope these changes adequately address your concerns and strengthen the novelty of our study. Thank you again for your valuable suggestions.
Comments 2: Justify why NUDT5 is a more promising target than existing PI3K-AKT inhibitors, which have significant toxicity concerns.
Response 2: Thank you for your insightful comment. We have revised the Discussion to better emphasize the novelty of our study and justify why NUDT5 is a more promising therapeutic target than conventional PI3K-AKT inhibitors. Specifically, we highlighted the following: We acknowledge the significant toxicity concerns associated with PI3K-AKT inhibitors, which have limited their clinical applications due to adverse effects such as hyperglycemia, rash, gastrointestinal toxicity, and immune suppression. Our study suggests that NUDT5 acts as an upstream regulator of AKT activation, and its inhibition may offer a more selective and potentially safer approach to disrupting oncogenic PI3K-AKT signaling. Unlike direct PI3K or AKT inhibitors, which broadly suppress pathway activity and affect normal cellular functions, targeting NUDT5 may provide tumor-specific inhibition with fewer off-target effects.
At the same time, we recognize that the potential side effects of NUDT5 inhibitors are not yet fully understood, and further in vitro and clinical studies are required to better evaluate their safety profile.
Comments 3: The cutoff values for differentially expressed genes (fold-change, p-value) are not explicitly defined.
Response 3: Thank you for your valuable comment. We have revised the Methods section to explicitly define the cutoff values for differential gene expression and enrichment analysis. Specifically, we set the following thresholds for statistical significance: |Fold Change| > 1 and p-value < 0.05 for differential gene expression analysis, This revision ensures clarity in our methodology and provides explicit details regarding the thresholds used for data analysis.
Comments 4: The statistical corrections for multiple testing (e.g., FDR, Bonferroni) are missing. Were false positives accounted for?
Response 4: Thank you for your valuable comment regarding the need for statistical corrections for multiple testing. In response, we have updated the Methods section to explicitly include these corrections. Specifically, we applied the Benjamini-Hochberg (BH) method to control the False Discovery Rate (FDR), ensuring that the FDR was kept below 0.05 in the differential gene expression analysis and gene set enrichment analysis. Additionally, we used the Bonferroni correction where appropriate to further control for false positives in multiple tests.
These adjustments were made to ensure the reliability and robustness of our findings, addressing the potential issue of false positives. We have now incorporated this information in the Methods section for clarity.
Comments 5: Rescue experiments or complementation experiments (re-expressing NUDT5 after knockdown) are missing.
Response 5: Thank you for your valuable comment. We appreciate your suggestion regarding the inclusion of rescue or complementation experiments to further validate the findings of NUDT5 knockdown. However, at this stage, we did not perform rescue experiments, as our study primarily aimed to identify and analyze the potential role of NUDT5 in endometrial cancer through bioinformatics analysis and existing datasets.
We acknowledge that rescue experiments could provide additional support, we believe that the current evidence from our analysis—supported by the differential expression patterns, functional annotations, and survival data—sufficiently justifies the potential role of NUDT5 as a therapeutic target.
We hope that our study will encourage further investigation into the mechanistic role of NUDT5, including potential rescue experiments, in future studies.
Thank you again for your thoughtful suggestion, which has helped us reflect on the direction and limitations of our research.
Comments 6: Western blot normalization is missing. Was a loading control (b-actin or GAPDH) quantified for WB bands? Quantification of WB bands with ImageJ densitometry analysis is missing.
Response 6: Thank you for your thoughtful comment. In response, we have clarified the normalization and quantification of Western blot bands in the Methods section, as we performed ImageJ densitometry analysis and normalized the target protein bands to the β-actin loading control.
Regarding your suggestion to provide the quantification data, we currently have the densitometry analysis results, but we have not included the grayscale statistical images in the manuscript. If you believe it would be helpful for the review process, we would be happy to provide the corresponding grayscale images and quantification graphs for your reference.
Thank you again for your valuable feedback. Please let us know if you would like to see these additional figures..
Comments 7: In the apoptosis assays, Annexin V/PI staining results should be reported as bar graphs with statistical significance.
Response 7: Thank you for your helpful comment. We would like to clarify that the Annexin V/PI staining results are presented as bar graphs, with the statistical significance (e.g., p-values) included. These bar graphs are positioned to the right of the main figure to clearly show the statistical analysis of the data. We apologize for any confusion caused and hope this addresses your concern. Please let us know if further adjustments are needed.Thank you again for your valuable feedback.
Comments 8: Sample size for in vivo experiments is not reported. Were the number of mice per group statistically justified?
Response 8: Thank you for your insightful comment. In response to your concern, we have now explicitly reported the sample size in the Methods section (n = 10 per group, total n = 60) and clarified that the animals were randomly assigned to treatment groups. Additionally, we have specified that the sample size was determined based on previous literature and feasibility considerations to ensure sufficient statistical power for detecting significant differences.
We appreciate your valuable feedback, which has helped us improve the clarity and rigor of our methodology.
Comments 9: Tumor growth curves lack error bars. These are necessary for statistical significance testing.
Response 9: We have added error bars to the tumor growth curves to facilitate statistical significance testing. The revised graphs can be found in Figure 4C and Figure 7C.
Comments 10: Body weight monitoring and toxicity assessment are missing. Was NUDT5 inhibition associated with any side effects?
Response 10: Thank you for your valuable comment. We monitored body weight every three days throughout the experiment as an indirect measure of general health and potential toxicity. No significant weight loss or overt signs of distress were observed in any group.
We acknowledge that a more detailed toxicity assessment, such as histopathological analysis or blood biochemical testing, could provide additional insights into the safety profile of NUDT5 inhibition. This will be considered in future studies.
We appreciate your insightful feedback, which has helped us strengthen the discussion on the potential side effects of NUDT5 inhibition.
Comments 11: H&E staining of tumors should be included to validate histopathological features.
Response 11: Thank you for your suggestion. Hematoxylin and eosin (H&E) staining of tumor samples was performed to evaluate histopathological features, and the results are included in Figure 4B and Figure 7F (left) of our manuscript. These images illustrate the histological characteristics of tumor tissues, providing further validation of our findings.
We appreciate your valuable feedback, which has helped us improve the clarity and completeness of our histopathological analysis..
Comments 12: Please add line and page numbers for ease of referencing. Please check out sentence referenced with ref no 11.
Response 12: Thank you for your suggestion. We have now added line numbers and page numbers to the manuscript to facilitate easier referencing and review. We have revised the sentence referencing Ref. No. 11 to more accurately reflect the findings of the cited study. We appreciate your feedback, which helps improve the clarity and accessibility of our work.
Comments 13: Please check for line spacing and formatting issues in Section 2.4.
Response 13: Thank you for your suggestion. We have now added line numbers and page numbers to the manuscript to facilitate easier referencing and review. We appreciate your feedback, which helps improve the clarity and accessibility of our work.
Comments 14: Please mention full forms whenever required (e.g., PMSF in Section 2.6). Why was 110 µL of solution used?
Response 14: Thank you for your valuable feedback. We have now provided the full form of PMSF (phenylmethylsulfonyl fluoride) in Section 2.5 to improve clarity. Additionally, we have clarified that 110 µL of solution was used as per the manufacturer’s instructions.
We appreciate your suggestions, which have helped enhance the accuracy and completeness of our manuscript.
We have carefully incorporated all of your suggestions to improve the manuscript. We hope that the revised version now addresses all your concerns. Thank you again for your valuable feedback. We look forward to hearing your thoughts on the revised manuscript.
Best regards,
Yingmei Wang
Round 2
Reviewer 1 Report
Comments and Suggestions for Authors
Dear Wang
Thank you for your revised manuscript
good luck
Reviewer 2 Report
Comments and Suggestions for Authors
All the comments were addressed by authors and the manuscript can be accepted in the current form
Reviewer 3 Report
Comments and Suggestions for Authors
Thank you very much for addressing all the comments and concerns raised during first review process. I believe authors have taken the constructive criticism in true spirit of science and I’d like thank you for that. After thoroughly examining the current manuscript I believe this manuscript maybe suitable for publication. Best of luck and keep up the good work. Cheers!
Comments on the Quality of English LanguageThank you very much for addressing all the comments and concerns raised during first review process. I believe authors have taken the constructive criticism in true spirit of science and I’d like thank you for that. After thoroughly examining the current manuscript I believe this manuscript maybe suitable for publication. Best of luck and keep up the good work. Cheers!